# FaceID-6M: A Large-Scale, Open-Source FaceID Customization Dataset

## Abstract

Due to the data-driven nature of current face identity (FaceID) customization methods, all state-of-the-art models rely on large-scale datasets containing millions of high-quality text-image pairs for training. However, none of these datasets are publicly available, which restricts transparency and hinders further advancements in the field.

To address this issue, in this paper, we collect and release FaceID-6M, the first large-scale, open-source FaceID dataset containing 6 million high-quality text-image pairs. Filtered from LAION-5B (Schuhmann et al., 2022), FaceID-6M undergoes a rigorous image and text filtering steps to ensure dataset quality, including resolution filtering to maintain high-quality images and faces, face filtering to remove images that lack human faces, and keyword-based strategy to retain descriptions containing human-related terms (e.g., nationality, professions and names). Through these cleaning processes, FaceID-6M provides a high-quality dataset optimized for training powerful FaceID customization models, facilitating advancements in the field by offering an open resource for research and development.

We conduct extensive experiments to show the effectiveness of our FaceID-6M, demonstrating that models trained on our FaceID-6M dataset achieve performance that is comparable to, and slightly better than currently available industrial models. Additionally, to support and advance research in the FaceID customization community, we make our code, datasets, and models fully publicly available.

## 1 Introduction

Face identity (FaceID) customization is an important task in image generation, in which users can write text prompts to adapt pre-trained text-to-image models to generate personalized facial images (Nichol et al., 2021; Ramesh et al., 2022; Saharia et al., 2022; Rombach et al., 2022; Gal et al., 2022; Kumari et al., 2023; Ruiz et al., 2023; Xu et al., 2024).

Existing approaches towards developing FaceID customization models are predominantly data-driven. This dataset typically consists of millions of real-world text-image pairs, such as 10M for IP-Adapter (Ye et al., 2023), 10M for InstantID (Wang et al., 2024) and 1.5M for PuLID (Guo et al., 2024). Then, a conditional diffusion model (Ho et al., 2020; Rombach et al., 2022; Peebles & Xie, 2023; Podell et al., 2023; Esser et al., 2024) is trained on this dataset, learning to reconstruct the original image while being conditioned on both the provided text description and the face within the input image. However, **none of these datasets are publicly accessible**. Although many studies have released high-performance FaceID models, they have not made their training code or datasets available to the broader research community, as a result, restricting transparency and hindering further advancements in the field.

To address this issue, in this paper, we collect and release FaceID-6M, the first large-scale, open-source faceID dataset containing 6 million high-quality text-image pairs. Filtered from LAION-5B (Schuhmann et al., 2022), which includes billions of diverse and publicly available text-image pairs, FaceID-6M undergoes a rigorous image and text filtering process to ensure dataset quality. For image filtering, we apply a pre-trained face detection model to remove images that lack human faces, contain more than three faces, have low resolution, or feature faces occupying less than 4% of the

total image area. For text filtering, we utilize a keyword-based strategy to retain descriptions containing human-related terms, including references to people (e.g., man), nationality (e.g., Chinese), ethnicity (e.g., East Asian), professions (e.g., engineer), and names (e.g., Donald Trump). Through these cleaning processes, FaceID-6M provides a high-quality dataset optimized for training powerful FaceID customization models, facilitating advancements in the field by offering an open resource for research and development.

FaceID-6M is a model-free FaceID customization dataset that can be utilized by any FaceID customization framework to train powerful FaceID customization models, e.g., IP-Aadapter (Ye et al., 2023), InstantID (Wang et al., 2024) and PuLID (Guo et al., 2024). During the training stage, we first leverage a pre-trained face detection model (e.g., Antelopev2 [1]) to extract the face within the image for one text-image training pair in our FaceID-6M dataset. Then, a diffusion model is forced to learn to recover the related image conditioned with the extracted face and related text as the input. Iterating this training process, the diffusion model can have the ability to generate the desired image based on the input personalized face and text.

To assess the effectiveness of FaceID-6M in training FaceID customization models, we conduct extensive experiments, including direct generation comparisons between FaceID-6M-trained models and existing industrial models, quantitative evaluations on two widely used test sets, COCO2017 (Lin et al., 2014) and Unsplash-50 (Gal et al., 2024), and human evaluations to further validate performance. The results show that models trained on our FaceID-6M achieve performance that is comparable to, and slightly better than, two widely used FaceID customization frameworks, InstantID (Wang et al., 2024) and IP-Adapter (Ye et al., 2023), in terms of preserving FaceID fidelity across all metrics. For example, on the COCO-2017 test set, the FaceID-6M-trained InstantID model achieved a slightly higher FaceID fidelity score of 0.63 (+0.04) compared to the score 0.59 of the official InstantID model. Similarly, in human evaluations, the FaceID-6M-trained InstantID model received an average score of 4.39 (+0.13), outperforming the official InstantID model's score of 4.26. The contribution of this work can be summarized as:

1. We collect FaceID-6M, the first large-scale, open-source FaceID customization dataset.

2. We make our code, datasets, and models publicly available to support and advance research in the FaceID customization community.

3. Models trained on our FaceID-6M dataset demonstrate performance that is comparable to, and better than currently available industrial models.

## 2 RELATED WORK

**Text-to-image Generation.** Text-to-image generation is the process of creating images from textual descriptions using pre-trained image generation models (Ramesh et al., 2021; Ding et al., 2021; 2022; Ramesh et al., 2022; Rombach et al., 2022; Saharia et al., 2022; Huang et al., 2023). These models are trained to map textual input to corresponding visual content, enabling them to generate images that align with the provided descriptions. Early text-to-image approaches typically followed a two-stage process: firstly, images are encoded into discrete tokens using an image encoder such as DARN(Gregor et al., 2014), PixelCNN(Van den Oord et al., 2016), PixelVAE (Gulrajani et al., 2016), or VQ-VAE (Van Den Oord et al., 2017). Then, a Transformer-based model (Vaswani, 2017) is trained to predict these image tokens based on the given text input. Recently, diffusion models (Song et al., 2020a;b; Nichol et al., 2021; Dhariwal & Nichol, 2021; Ramesh et al., 2022; Saharia et al., 2022; Rombach et al., 2022; Balaji et al., 2022; Huang et al., 2023) have emerged as the new state-of-the-art approach for image generation, introducing more advanced techniques for text-to-image synthesis. Based on diffusion frameworks, firstly, a pre-trained large language model, such as T5 (Raffel et al., 2020), is used to encode the input text into embeddings. Then, the transformed text embeddings are used as conditional inputs for the diffusion model, guiding it to generate high-quality images based on the provided textual descriptions (Nichol et al., 2021).

**FaceID Customization** FaceID customization involves adapting a text-to-image generation model to better recognize and generate facial features and attributes unique to individual users (Valevski et al., 2023; Ye et al., 2023; Yuan et al., 2023; Chen et al., 2024; Wang et al., 2024; Xiao et al.,

---

[1]https://github.com/deepinsight/insightface

2024; Li et al., 2024; Peng et al., 2024). Most existing approaches are data-driven, begining with the collection of a large-scale dataset, such as 10M for IP-Adapter (Ye et al., 2023), 10M for InstantID (Wang et al., 2024) and 1.5M for PuLID (Guo et al., 2024). Then, a conditional diffusion model (Ho et al., 2020; Rombach et al., 2022; Peebles & Xie, 2023; Podell et al., 2023; Esser et al., 2024) is trained on this dataset, such as Face0 (Valevski et al., 2023) replaces the last three text tokens with a projected face embedding in the CLIP (Radford et al., 2021) space, using the combined embedding to guide the diffusion process. IP-Adapter, InstantID (Wang et al., 2024) and PuLID (Guo et al., 2024) utilize FaceID embeddings from a face recognition model instead of CLIP image embeddings, ensuring a more stable and consistent identity representation throughout the generation process. However, none of these studies have made their training datasets publicly available to the FaceID customization community, restricting and hindering further advancements in this filed. To address this gap, in this paper, we introduce FaceID-6M, a large-scale, open-source FaceID customization dataset. FaceID-6M is constructed through a rigorous image and text filtering process applied to LAION-5B (Schuhmann et al., 2022), a diverse and publicly available text-image dataset, ensuring its high quality for training advanced FaceID customization models.

## 3 DATASET CONSTRUCTION: FACEID-6M

In this section, we detail the construction process of our FaceID-6M dataset, which comprises four main components: **(1) Text-Image Pairs Collecting:** Gathering a large number of text-image pairs as the foundational dataset, which will undergo further filtering to ensure relevance for the FaceID customization task; **(2) Language Filtering:** Filtering out non-English text-image pairs, as many FaceID models are designed with English as the primary, and non-English text may not be accurately understood or processed by the model; **(3) Image Filtering:** Removing low-quality, inappropriate, or irrelevant images that do not meet the requirements of FaceID-related tasks, such as images without faces or those containing blurred or occluded faces; and **(4) Text Filtering:** Removing pairs whose text descriptions that are irrelevant or misleading for FaceID applications, such as captions that do not provide any meaningful information about the person in the corresponding image.

### 3.1 TEXT-IMAGE PAIRS COLLECTING

To construct a large-scale, high-quality FaceID dataset, we utilize LAION-5B (Schuhmann et al., 2022), a publicly available dataset containing billions of text-image pairs. LAION-5B is an ideal foundation for two key reasons: (1) Extensive Data Size: its vast scale provides a broad spectrum of data, crucial for training robust image generation models. The dataset's diversity, spanning various domains and textual descriptions, enables the model to learn from a wide array of contexts; (2) Public Accessibility: as an openly available dataset, LAION-5B allows readers to reproduce our results and make further advancements without the need for extensive proprietary data collection

### 3.2 LANGUAGE FILTERING

In this paper, we specifically select the English subset of LAION-5B, which consists of approximately 2 billion English text-image pairs. This choice is driven by two key reasons: (1) The quality of textual annotations in LAION varies across different languages. English captions are generally more structured, detailed, and reliable, as many online sources predominantly use English for annotations; and (2) The FaceID customization task relies on powerful vision-language models to establish strong text-image alignment. English-based models typically outperform their counterparts in other languages, ensuring better filtering accuracy and enhanced performance in FaceID customization model training. Figure 3 shows statistics for the English subset of LAION-5B.

### 3.3 IMAGE FILTERING

As revealed in Figure 3, even limited to the English language, the raw LAION-5B dataset contains a significant amount of noisy data, making it unsuitable for directly training FaceID customization models: (1) a large proportion of low-resolution images, such as 37.7% falling between 128 to 256 pixels in height and 35.4% between 256 to 512 pixels in width. These low-resolution images fail to provide adequate facial details, such as expressions and facial textures, which are crucial for identity preservation; (2) there is no guarantee of a sufficient number of high-quality human face

images, which are crucial for the FaceID customization task. To effectively filter and extract relevant samples from LAION, we leverage the following criteria:

**(1) Face Detection.** To ensure every image in the training dataset contains faces, we utilize a pre-trained face detection model, Antelopev2[2], to identify face IDs and filter out images that either lack faces or contain more than three faces. The limit of three faces per image is set because when there are multiple faces, the model may struggle to determine which face to prioritize, resulting in a mix of facial features from different individuals.

**(2) Minimum Face Size Constraints.** We set that the face must occupy at least 4% of the image area. This ensures that a sufficient number of pixels are contained into facial features, allowing the FaceID customization model to capture essential details such as expressions and facial textures, which are crucial for accurate identity preservation. The 4% threshold was established through iterative refinements and validated through both human and model evaluations.

**(3) Resolution Constraints.** To guarantee adequate facial details for the FaceID customization task, we exclude images with a resolution lower than 512 pixels in either height or width. This strategy enhances the FaceID customization model's ability to extract meaningful features, such as subtle expressions and facial textures, ultimately improving learning effectiveness and overall performance.

## 3.4 TEXT FILTERING

Even after selecting high-quality and human-related images, the associated textual descriptions in LAION may contain irrelevant or misleading information, such as captions that do not provide any meaningful information about the person depicted in the corresponding image. To refine the associated text, we employ a keyword-based filtering strategy, selecting samples whose descriptions include one or more of the following five categories of relevant terms:

1. Terms that explicitly denote individuals (e.g., man, woman, sir, lady).
2. Terms indicating nationality (e.g., Chinese, Korean).
3. Terms denoting ethnicity or racial identity (e.g., Native American, East Asian).
4. Terms related to professions or occupations (e.g., student, teacher, engineer).
5. Terms that potentially indicate a person's name (e.g., Donald Trump, Beckham).

To determine specific terms for categories 1 to 4, we leverage GPT-4o (Hurst et al., 2024), which has been trained on vast textual data, enabling it to generate an extensive and diverse list of relevant terms. Take the category 4 "Terms related to professions or occupations (e.g., student, teacher, engineer)" as an example, we use an iterative approach with 10 iterations. In each iteration, we prompt GPT-4o with a different temperature setting to generate 100 profession-related terms. After completing all 10 iterations, we consolidate the 1,000 generated terms, remove duplicates, and finalize a refined list of keywords for that category. During the filtering process, each LAION text is checked against this keyword list. If a text contains at least one of these keywords, it is retained; otherwise, it is discarded.

For identifying specific terms in category 5, GPT-4o is not suitable, as the number of possible names is virtually unlimited, making it impossible to generate a comprehensive predefined list. Instead, determining whether a word is a person's name should rely on analyzing its position within a sentence and evaluating its contextual relevance rather than relying solely on a static list. To achieve this, we utilize an NER-based filtering strategy for LAION text. Specifically, for each text in LAION, we utilize spaCy [3] to extract PERSON entities, which include references to both real and fictional individuals. Any LAION text that does not contain a PERSON entity is filtered out.

Following the above cleaning process, we ensure that the collected images contain clear, adequately sized faces, and that the accompanying text provides meaningful descriptions of the person in the image. Figure 2 presents examples from our cleaned FaceID-6M dataset.

---

[2] https://github.com/deepinsight/insightface

[3] A commonly used NER toolkit: https://spacy.io/api/entityrecognizer.

# 4 FACEID CUSTOMIZATION BASED ON FACEID-6M

In this section, we first detail the preliminaries of Diffusion Models in Section 4.1. Then, we delve into the training and inference processes of our FaceID-6M-based FaceID customization models using the widely adopted FaceID framework, IP-Adapter (Ye et al., 2023), in Section 4.2.

## 4.1 PRELIMINARIES: DIFFUSION MODELS

Diffusion models (Ho et al., 2020; Rombach et al., 2022; Podell et al., 2023) are a class of generative models that have gained prominence for their ability to generate high-quality, realistic data, such as images, by simulating a gradual process of transforming random noise into structured data. Specifically, during each training process, noise $\epsilon$ is sampled and added to the input image $x_0$ based on a noise schedule (i.e., Gaussian noise). This process yields a noisy sample $x_t$ at timestep $t$:

$$x_t = \alpha_t x + \sigma_t \epsilon, \epsilon \sim \mathcal{N}(0, \mathbf{I}) \tag{1}$$

where $\alpha_t$ and $\sigma_t$ are the coefficients of the adding noise process, essentially representing the noise schedule. Then the diffusion model $\epsilon_\theta$ is forced to predict the normally-distributed noise $\epsilon$ with current added noisy $x_t$, time step $t$, and condition information $C$, where commonly $C$ represents the embedded text prompt. For optimization process:

$$\mathcal{L}(\theta) = \mathbb{E}_{x_0, C, t, \epsilon \sim \mathcal{N}(0, \mathbf{I})} \|\epsilon - \epsilon_\theta(x_t, C, t)\|^2 \tag{2}$$

where $t \in [0, T]$ is the sampled diffusion step.

The inference stage begins with a sample of pure noise (Gaussian noise), represented by $x_T$, where $T$ is a predefined number of timesteps. This random noise is the initial state, representing a completely unstructured and meaningless input. Then, for each timestep $t$, the model takes the noisy image $x_t$ at step $t$ as the input, and incorporates the text prompt as the condition $C$ to predict the clean image or the noise that should be removed to get closer to the final clean image $x_0$. The predicted noise $\epsilon_\theta$ is then used to update the noisy image, denoising it step by step:

$$x_{t-1} = \alpha_t x_t - \sigma_t \epsilon_\theta(x_t, C, t) \tag{3}$$

where $\alpha_t$ and $\sigma_t$ are two coefficients controlling the denoising process. Finally, over several timesteps $T$, the noise is gradually removed, resulting in a high-quality image.

## 4.2 FACEID-6M BASED FACEID CUSTOMIZATION

FaceID-6M is a model-free FaceID customization dataset that can be utilized by any FaceID customization framework to train powerful FaceID customization models. Here, we employ the widely used FaceID customization framework, IP-Adapter (Ye et al., 2023), to illustrate the training process of FaceID customization models using our FaceID-6M dataset, followed by the inference stage.

### 4.2.1 TRAINING STAGE

During the training stage, we utilize a distinct decoupled cross-attention mechanism to embed image features through several extra cross-attention layers, while keeping the other model parameters intact. Specifically, in original diffusion models, the text features from the CLIP (Radford et al., 2021) or T5 (Raffel et al., 2020) text encoder are incorporated into the model by inputting them into the cross-attention layers. Given the latent image features $Z$ and the text features $C_{text}$, the output of cross-attention $Z^{'}$ can be expressed as:

$$Z^{'} = \text{Attention}(Q, K, V) = \text{Softmax}(\frac{QK^T}{\sqrt{d}})V, Q = ZW_q, K = C_{text}W_k, V = C_{text}W_v \tag{4}$$

where $Q$, $K$, and $V$ represent the query, key, and value matrices in the attention operation, respectively, while $W_q$, $W_k$, and $W_v$ are the weight matrices of the learnable layers.

To further incorporate face ID, IP-Adapter (Ye et al., 2023) adds a new cross-attention layer for each cross-attention layer in the original diffusion model. Similarly, given the face ID features $C_{id}$, the

output of ID cross-attention $Z^{''}$ is:

$$Z^{''} = \text{Attention}(Q, K^{'}, V^{'}) = \text{Softmax}(\frac{Q(K^{'})^T}{\sqrt{d}})V^{'}, Q^{'} = ZW_q^{'}, K^{'} = C_{id}W_k^{'}, V^{'} = C_{id}W_v^{'}$$

(5)

where $Q^{'}$, $K^{'}$, and $V^{'}$ represent the query, key, and value matrices in the attention operation, respectively, while $W_q^{'}$, $W_k^{'}$, and $W_v^{'}$ are the weight matrices of the learnable layers.

The text cross-attention $Z^{'}$ and the ID cross-attention $Z^{''}$ are added, resulting the final decoupled attention $Z^{final}$:

$$\begin{aligned} Z^{final} = Z^{'} + \lambda \cdot Z^{''} &= \text{Attention}(Q, K, V) + \lambda \cdot \text{Attention}(Q, K^{'}, V^{'}) \\ &= \text{Softmax}(\frac{QK^T}{\sqrt{d}})V + \lambda \cdot \text{Softmax}(\frac{Q(K^{'})^T}{\sqrt{d}})V^{'} \end{aligned}$$

(6)

where $\lambda$ is a weight factor.

During the training stage, IP-Adapter only optimizes the related linear layers within the decoupled cross-attention while keeping the parameters of the diffusion model fixed:

$$\mathcal{L}_{IP}(\theta) = \mathbb{E}_{x_0, C_{text}, C_{id}, t, \epsilon \sim \mathcal{N}(0, \mathbf{I})} \|\epsilon - \epsilon_\theta(x_t, C_{text}, C_{id}, t)\|^2$$

(7)

### 4.2.2 INFERENCE STAGE

The inference process follows the same approach as the diffusion models outlined in Section 4.1. It begins with a sample of Gaussian noise, represented by $x_T$, where $T$ is a predefined number of timesteps. This initial state, composed entirely of unstructured noise, serves as the starting point, representing a meaningless input image. At each timestep $t$, the model takes the noisy image $x_t$ as the input and utilizes the text prompt condition $C_{text}$ and the input face condition $C_{id}$ to predict the clean image or the noise that should be removed, progressively refining the image towards the final clean output $x_0$. The predicted noise $\epsilon_\theta$ is then used to update the noisy image, denoising step by step:

$$x_{t-1} = \alpha_t x_t - \sigma_t \epsilon_\theta(x_t, C_{text}, C_{id}, t)$$

(8)

where $\alpha_t$ and $\sigma_t$ are two coefficients controlling the denoising process. Over several timesteps $T$, the noise is gradually removed, ultimately producing a customized, clean image.

| Model | COCO-2017 | | | Unsplash-50 | | |
| --- | --- | --- | --- | --- | --- | --- |
| | Face Sim↑ | CLIP-T↑ | CLIP-I↑ | Face Sim↑ | CLIP-T↑ | CLIP-I↑ |
| IP-Adapter (Ye et al., 2023) | 0.52 | 0.46 | 0.69 | 0.57 | 0.24 | 0.61 |
| InstantID (Wang et al., 2024) | 0.59 | 0.53 | 0.72 | 0.61 | 0.27 | 0.68 |
| *Ours (Fine-tuned on Laion-FaceID)* | | | | | | |
| IP-Adapter + FaceID-6M | 0.55 | 0.48 | 0.70 | 0.59 | 0.24 | 0.62 |
| InstantID + FaceID-6M | **0.63 (+0.04)** | **0.54 (+0.01)** | **0.73 (+0.01)** | **0.62 (+0.01)** | **0.28 (+0.01)** | **0.70 (+0.02)** |

Table 1: Quantitative results of different FaceID customization models on COCO2017 and Unsplash-50, and we highlight the highest score in bold.

## 5 EXPERIMENTS

To assess the effectiveness of FaceID-6M, we train the state-of-the-art FaceID customization model, InstantID (Wang et al., 2024), as well as its original version, IP-Adapter (Ye et al., 2023), using our constructed FaceID-6M dataset.

### 5.1 MAIN RESULTS

In this section, we perform experiments to assess the FaceID fidelity of models trained on our custom FaceID-6M dataset. For clarity, we designate the official InstantID model as "InstantID" and the

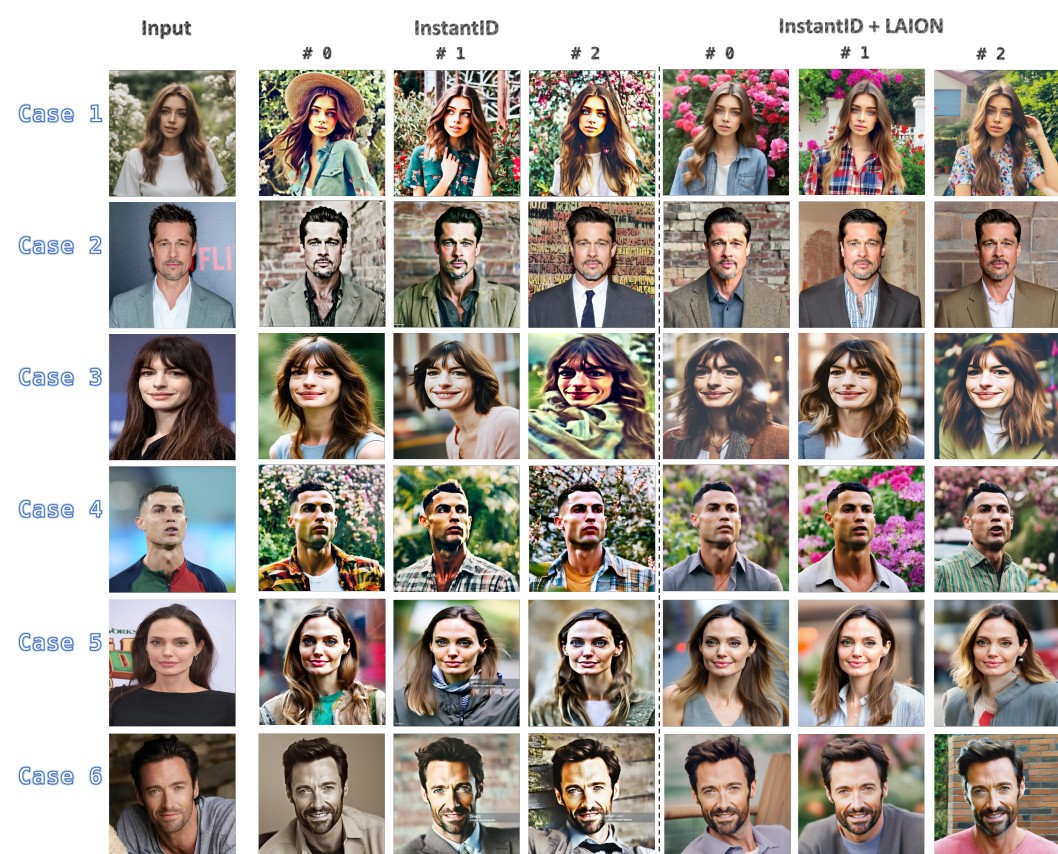

Figure 1: The results demonstrate the performance of FaceID customization models in maintaining FaceID fidelity. For models, "InstantID" refers to the official InstantID model, while "InstantID + FaceID-6M" represents the model further fine-tuned on our FaceID-6M dataset. These results indicate that the model trained on our constructed FaceID-6M dataset achieves comparable performance to the official InstantID model in preserving FaceID fidelity.

| Model | COCO-2017 | | | Unsplash-50 | | | Avg |
|---|---|---|---|---|---|---|---|
| | Face Sim↑ | CLIP-T↑ | CLIP-I↑ | Face Sim↑ | CLIP-T↑ | CLIP-I↑ | |
| InstantID + FaceID-6M | 0.63 | 0.54 | 0.73 | 0.62 | 0.28 | 0.70 | 0.58 |
| *Image Filtering* | | | | | | | |
| *w/o* Face Detection | 0.17 (-0.46) | 0.47 (-0.07) | 0.68 (-0.05) | 0.09 (-0.53) | 0.23 (-0.05) | 0.64 (-0.06) | 0.38 (-0.20) |
| *w/o* Resolution Constraints | 0.49 (-0.14) | 0.52 (-0.02) | 0.67 (-0.06) | 0.51 (-0.11) | 0.25 (-0.03) | 0.61 (-0.09) | 0.51 (-0.07) |
| *w/o* Minimum Face Size Constraints | 0.60 (-0.03) | 0.52 (-0.02) | 0.71 (-0.02) | 0.57 (-0.05) | 0.26 (-0.02) | 0.67 (-0.03) | 0.56 (-0.02) |
| *Text Filtering* | | | | | | | |
| *w/o* Including Individual Terms | 0.51 (-0.12) | 0.36 (-0.18) | 0.49 (-0.24) | 0.47 (-0.15) | 0.11 (-0.17) | 0.55 (-0.15) | 0.42 (-0.16) |
| *w/o* Including Nationality Terms | 0.62 (-0.01) | 0.53 (-0.01) | 0.72 (-0.01) | 0.60 (-0.02) | 0.26 (-0.02) | 0.67 (-0.03) | 0.57 (-0.01) |
| *w/o* Including Ethnicity Terms | 0.62 (-0.01) | 0.54 (-0.00) | 0.71 (-0.02) | 0.61 (-0.01) | 0.25 (-0.03) | 0.68 (-0.02) | 0.57 (-0.01) |
| *w/o* Including Occupation Terms | 0.60 (-0.03) | 0.53 (-0.01) | 0.69 (-0.04) | 0.57 (-0.05) | 0.26 (-0.02) | 0.68 (-0.02) | 0.56 (-0.02) |
| *w/o* Including Person's Name | 0.62 (-0.01) | 0.52 (-0.02) | 0.73 (-0.00) | 0.60 (-0.02) | 0.25 (-0.03) | 0.69 (-0.01) | 0.57 (-0.01) |

Table 2: Results of different FaceID customization models on COCO2017 and Unsplash-50 by leaving out one filtering strategy.

model trained on our FaceID-6M dataset as "InstantID + FaceID-6M." The results below illustrate their performance in FaceID fidelity.

### 5.1.1 FACEID FIDELITY

Figure 1 presents the performance of FaceID customization models in preserving FaceID fidelity. Based on these results, we can infer that the model trained on our FaceID-6M dataset achieves a

| Model | Prompt Alignment | | | FaceID Fidelity | | | Image Quality | | |
|---|---|---|---|---|---|---|---|---|---|
| | *Min* | *Max* | *Avg* | *Min* | *Max* | *Avg* | *Min* | *Max* | *Avg* |
| IP-Adapter Ye et al. (2023) | 1 | 3 | 1.8 | 1 | 3 | 2.17 | 2 | 4 | 2.94 |
| InstantID Wang et al. (2024) | 2 | 4 | 3.1 | 3 | 5 | 3.9 | 3 | 5 | 4.26 |
| *Ours (Fine-tuned on Laion-FaceID)* | | | | | | | | | |
| IP-Adapter + FaceID-6M | 1 | 3 | 1.83 | 1 | 3 | 2.19 | 2 | 4 | 2.95 |
| InstantID + FaceID-6M | **2** | **4** | **3.14** | **3** | **5** | **4.08** | **3** | **5** | **4.39** |

Table 3: Human evaluations of different FaceID customization models based on three criteria: (1) Prompt Alignment, (2) FaceID Fidelity and (3) Image Quality, and we highlight the highest score in bold.

level of performance comparable to the official InstantID model in maintaining FaceID fidelity. For example, in case 2 and case 3, both the official InstantID model and the FaceID-6M-trained model effectively generate the intended images based on the input. This clearly highlights the effectiveness of our FaceID-6M dataset in training robust FaceID customization models.

## 5.2 QUANTITATIVE RESULTS

To more effectively evaluate the effectiveness of our FaceID-6M dataset, we conduct quantitative experiments on two test sets: COCO2017 (Lin et al., 2014) and Unsplash-50 (Gal et al., 2024).

COCO2017 (Lin et al., 2014) consists of 5,000 images with captions suitable for quantitative evaluation. However, since our primary focus is on evaluating models' ability to maintain FaceID fidelity, many samples in COCO2017 are not relevant to our objective. Therefore, in this study, we manually selected 500 valid text-image pairs to quantitatively assess the models' performance in preserving FaceID fidelity.

Unsplash-50 (Gal et al., 2024) contains 50 text-image pairs, which serve as an additional benchmark for evaluating FaceID fidelity retention in generated images.

For evaluation metrics, we use the following: (1) Face Sim, which calculates the FaceID cosine similarity between the input face and the face extracted from the generated image, providing a direct estimate of the difference between the generated and input faces. (2) CLIP-T (Radford et al., 2021), which evaluates the model's ability to follow prompts; and (3) CLIP-I (Radford et al., 2021), which measures the CLIP image similarity between the original image and the image after FaceID insertion.

The results are presented in Table 1. From these findings, we observe that fine-tuning on our FaceID-6M dataset leads to slight improvements across all evaluation metrics. For example, the Face-Sim score on the COCO2017 dataset increases from 0.59 (LAION) to 0.63 (InstantID + FaceID-6M), while the CLIP-I metric on the Unsplash-50 dataset improves from 0.68 (LAION) to 0.70 (InstantID + FaceID-6M). These results further confirm the effectiveness of our FaceID-6M dataset in enhancing the model's ability to preserve the input face's identity while generating images that align with user descriptions.

## 5.3 FILTERING STRATEGY IMPACT

In this paper, we introduce three image filtering strategies (in Section 3.3, "Image Filtering") and five text filtering strategies (in Section 3.4, "Text Filtering"). To evaluate their effects, we train the InstantID model on eight dataset variants, as described in Section 5.2, "Quantitative Results". Each variant is created by leaving out one filtering strategy, and we then compare these results with our fully filtered dataset to measure the impact of each strategy.

Results are shown in Table 2, and from these results, we can observe that: (1) every filtering strategy plays an important role in training an effective FaceID customization model, though their impacts differ. For example, on the COCO-2017 dataset, removing "Face Detection" lowers the FaceSim score by 0.46, while removing "Including Individual Terms" reduces the CLIP-T score by 0.18; and (2) the strategy that discards images without faces has the strongest effect on model performance.

Specifically, leaving out "Face Detection" decreases the average score by 0.20, as the model struggles to learn effectively in a noisy setting when face information is absent.

## 5.4 Scaling Results

To evaluate the impact of dataset size on model performance and optimize the trade-off between performance and training cost, we conduct scaling experiments by sampling subsets of different sizes from FaceID-6M. The sampled dataset sizes include: (1) 1K, (2) 10K, (3) 100K, (4) 1M, (5) 2M, (6) 4M, and (7) the full dataset (6M). For the experimental setup, we utilize the InstantID (Wang et al., 2024) FaceID customization framework and adhere to the configurations used in the previous quantitative evaluations. The trained models are tested on the COCO2017 (Lin et al., 2014) test set, with Face Sim, CLIP-T, and CLIP-I as the evaluation metrics.

The results, presented in Figure 4, demonstrate a clear correlation between training dataset size and the performance of FaceID customization models. For example, the Face Sim score increased from 0.38 with 2M training data, to 0.51 with 4M, and further improved to 0.63 when using 6M data. These results underscore the significant contribution of our FaceID-6M dataset in advancing FaceID customization research, highlighting its importance in driving improvements in the field.

## 5.5 Human Evaluations

While automated evaluations, as conducted above, effectively measure objective aspects like FaceID fidelity and prompt adherence, they fall short in assessing subjective qualities, like aesthetic appeal, and perceptual consistency.

To address this limitation, we conduct a user study to gather human evaluations on image quality and identity preservation. Participants are asked to rate 200 generated images based on three criteria: (1) **Prompt Alignment:** assesses how well the generated image corresponds to the given textual description, (2) **FaceID Fidelity:** evaluates whether the generated image accurately preserves the identity of the input face, and (3) **Image Quality:** measures the overall visual quality of the generated image. Participants rate each criterion on a scale from 0 to 5, where 0 represents the lowest quality and 5 indicates the highest quality.

The results are shown in Table 3. From these results, we observe a slight improvement across all three criteria after fine-tuning on our FaceID-6M dataset: (1) Prompt Alignment, the average score increased from 3.1 for the InstantID model to 3.14 for the InstantID + FaceID-6M model. (2) FaceID Fidelity, the scores improved from 2.17 (IP-Adapter) to 2.19 (IP-Adapter + FaceID-6M) and from 3.9 (InstantID) to 4.08 (InstantID + FaceID-6M). (3) Image Quality, the score increased from 4.26 (InstantID) to 4.39 (InstantID + FaceID-6M). These results further demonstrate the benefits of fine-tuning with our FaceID-6M dataset in enhancing identity preservation, adherence to prompts, and overall image quality.

## 6 Conclusion

In this paper, we collect and release FaceID-6M, a large-scale, open-source dataset containing 6 million high-quality text-image pairs. FaceID-6M is filtered from LAION-5B, which includes billions of diverse and publicly available text-image pairs, and undergoes a rigorous image and text filtering process to ensure dataset quality. Specifically, for image filtering, we apply a pre-trained face detection model to remove images that lack human faces, contain more than three faces, have low resolution, or feature faces occupying less than 4% of the total image area. For text filtering, we use a keyword-based strategy to retain descriptions containing human-related terms, including references to people (e.g., man), nationality (e.g., Chinese), ethnicity (e.g., East Asian), professions (e.g., engineer), and names (e.g., Donald Trump). Through these cleaning processes, FaceID-6M provides a high-quality dataset optimized for training powerful FaceID customization models, facilitating advancements in the field by offering an open resource for research and development.

We conduct extensive experiments to show the effectiveness of our FaceID-6M, demonstrating that models trained on our FaceID-6M dataset achieve performance that is comparable to, and slightly better than currently available industrial models. Additionally, to support and advance research in the FaceID customization community, we make our code, datasets, and models fully publicly available.

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

# A    APPENDIX

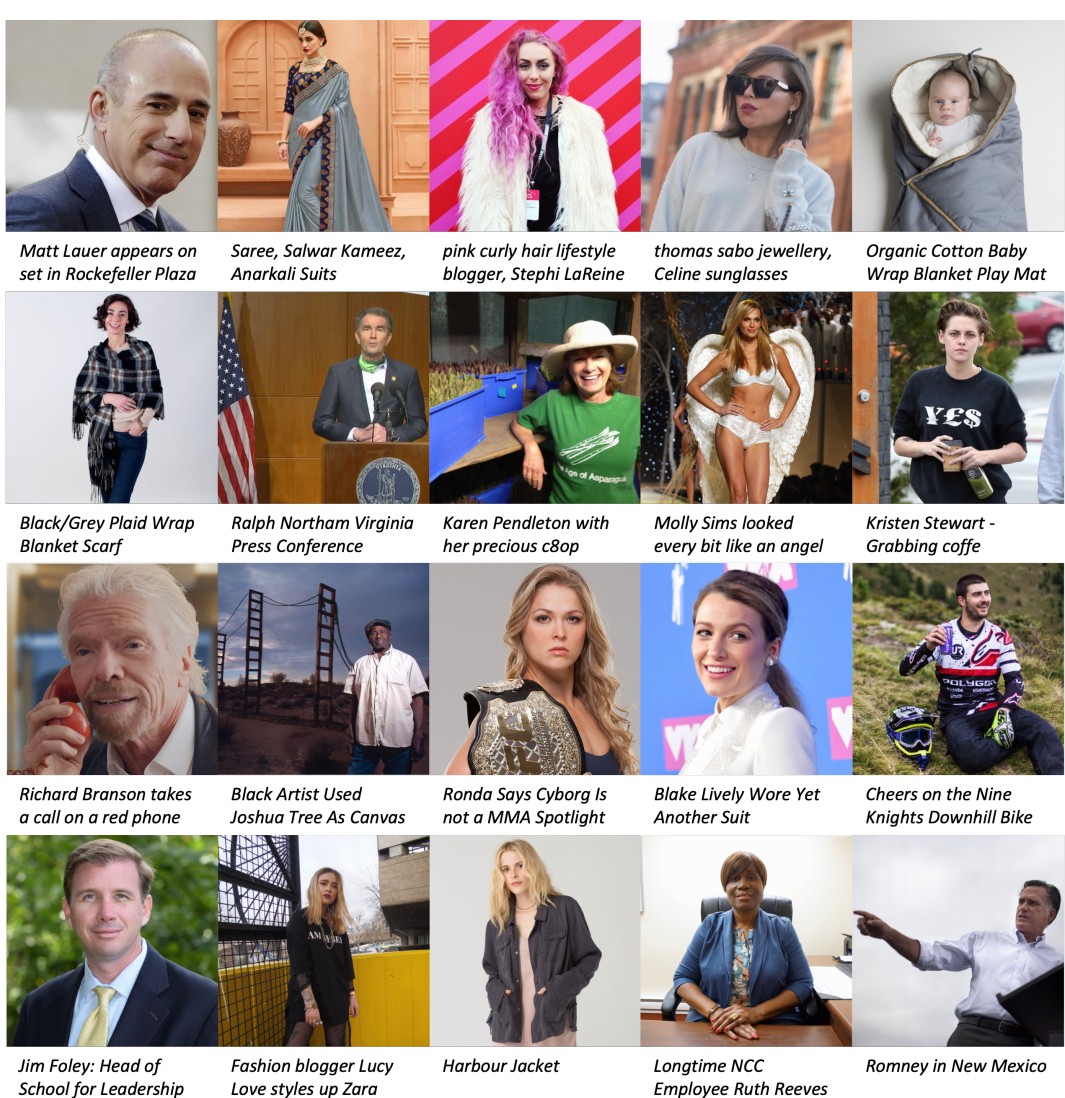

Figure 2: Images sampled from our constructed FaceID-6M dataset are presented.

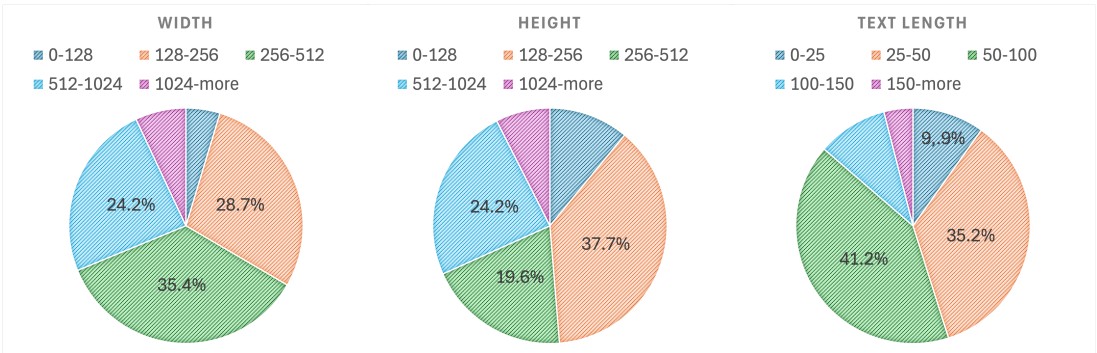

Figure 3: Statistics for the English subset of LAION-5B, presenting (1) image width, (2) image height, and (3) text length from left to right.

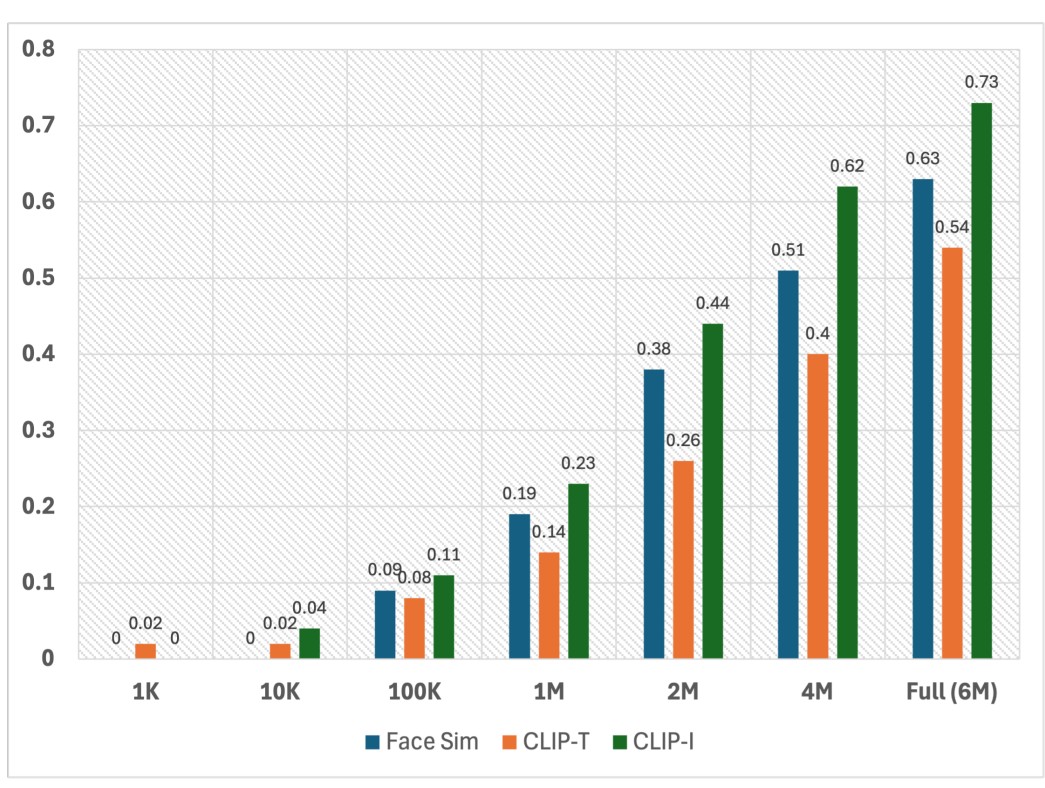

Figure 4: Scaling results by sampling subsets of different sizes from FaceID-6M: (1) 1K, (2) 10K, (3) 100K, (4) 1M, (5) 2M, (6) 4M, and (7) the full dataset (6M).

