# OpenReview forum: "FaceID-6M: A Large-Scale, Open-Source FaceID Customization Dataset"
_ICLR.cc/2026/Conference — Submitted to ICLR 2026_

### Official Review · Reviewer_NiYi · 2025-10-23

**Soundness:** 3
**Presentation:** 3
**Contribution:** 3
**Rating:** 6
**Confidence:** 4

**Summary:**

FaceID-6M presents a 6-million–pair, open-source dataset for FaceID customization built from LAION-5B via English-only selection, face-centric image filtering (face present, ≤3 faces, ≥512 px, ≥4% face area), and human-related text filtering using keyword lists and NER for names.

The dataset is positioned as model-agnostic and used to fine-tune IP-Adapter and InstantID; evaluations on COCO-2017 and Unsplash-50 report small but consistent gains in identity fidelity and CLIP metrics (e.g., FaceSim 0.59→0.63 on COCO) and modest improvements in human ratings. Ablations show that each filtering step matters (the largest drop occurs when omitting face detection), and scaling experiments indicate monotonic benefits up to the full 6M set. Claimed contributions are: (i) the first large-scale public FaceID customization dataset; (ii) full release of code/data/models to enable reproducibility; and (iii) parity or slight improvements over industrial models trained on private data.

**Strengths:**

The paper’s main strength is its decision to frame originality around removing a key limitation in FaceID personalization, closed proprietary corpora by releasing a large-scale, model-agnostic dataset with a transparent construction protocol; the contribution is practical rather than algorithmic but still creative in how standard components (face detection, resolution/face-area thresholds, keyword and NER filtering) are combined into a coherent pipeline optimized for identity-conditioned generation.

Targeted experiments beyond headline metrics support methodological quality: the paper includes quantitative benchmarks on two test sets, a human study, ablations that isolate the effect of each filtering stage (with face detection the most critical), and a scaling analysis that links performance to corpus size together indicating that the curation choices, not only dataset scale, drive the observed gain. Clarity is strong: the dataset construction, training/inference setup (using IP-Adapter as a worked example), and evaluation criteria are explained in self-contained sections with standard diffusion formulations, making the work reproducible for different FaceID frameworks.

In terms of significance, an open 6M-pair FaceID dataset, along with code/models, significantly lowers the barrier for academic and industrial research, enabling parity or slight improvements over widely used closed models and providing a common testbed for future algorithmic advances in identity fidelity and prompt alignment.

**Weaknesses:**

The paper’s central weakness is that its novelty is primarily curatorial: the pipeline (English-only subset selection, face detection, resolution and face-area thresholds, keyword and NER filtering) closely follows standard LAION-style refinement practices without introducing a new data governance protocol, deduplication scheme, or identity annotation standard; the specific thresholds (≤3 faces, face area ≥4%, min side ≥512 px) are asserted but not theoretically or empirically justified with sensitivity analyses, which weakens generality claims.

Evaluation is narrow and statistically underpowered. The main quantitative results utilize a hand-selected 500-pair subset of COCO-2017 and a 50-pair Unsplash set, neither of which is designed for identity-conditioned personalization. Therefore, the small gains (+0.04 FaceSim, +0.01–0.02 CLIP) may not be reliable or generalizable.

Human evaluation (200 images) lacks inter-rater reliability statistics and sampling protocol details. The ablation study is useful but incomplete. It isolates each filtering step by removal (Table 2) and shows that face detection has the most significant impact. Yet, it does not examine interactions (e.g., low-resolution × small-face) or trade-offs in text-filter precision and recall between keyword and NER choices. A factorial ablation and error audit (false positives/negatives in each filter) would make the quality claim more convincing.

Claims of “high-quality” data and “advancing the field” are insufficiently evidenced by dataset forensics. There is no audit of residual noise (NSFW, cartoons, synthetic faces), identity duplication, caption misalignment, or demographic composition.

Ethics, privacy, and safety are underdeveloped. The paper proposes releasing six million face-containing images scraped via LAION without discussing consent, minors, jurisdictional compliance, or anti-misuse safeguards. The text explicitly relies on name detection and human-attribute keywords, which can heighten the re-identification risk, yet no mitigation or governance measures are described.

**Questions:**

1. Curation thresholds and pipeline design. What empirical basis supports the ≤3 faces, ≥4% face-area, and ≥512-px thresholds? Please provide sweeps and Pareto curves showing identity fidelity vs. diversity when varying these cutoffs, not just the single operating point.

2. Dataset forensics and quality audits
   • Duplicates and identity fragmentation: Have you measured near-duplicate rates and identity fragmentation (same person across multiple captions/splits)? Please provide an embedding-based dedup pipeline, threshold calibration, and manual validation protocol.
   • Content hygiene: What fraction of the corpus is NSFW, synthetic/AI-generated, cartoons, or faces with heavy occlusion? Provide screened percentages and removal rules.

3. Evaluation scope and statistical rigor
   • Test sets: Why were a hand-selected 500-pair COCO subset and Unsplash-50 chosen for identity-conditioned personalization, and how do results transfer to face-centric benchmarks? Please add evaluations on established face corpora (e.g., FFHQ/CelebA-HQ subsets with text prompts) and more diverse prompts.
   • Scaling controls: In Fig. 4, were training steps/epochs and compute held constant across dataset sizes? Please report compute budgets to separate “more data” from “more training” effects【】.

4. Metrics and task framing
   • Biometric relevance: Complement FaceSim/CLIP with biometric-style metrics: verification EER at fixed FAR and closed/open-set identification Rank-k, using a held-out gallery, to substantiate “FaceID fidelity” claims.
   • Open-vs-closed conditions: Evaluate robustness under pose/lighting/expression shifts and cross-camera settings; report degradation curves to show practical personalization value.

---

### Official Review · Reviewer_c4sY · 2025-10-25

**Soundness:** 2
**Presentation:** 2
**Contribution:** 1
**Rating:** 2
**Confidence:** 2

**Summary:**

The paper introduces FaceID-6M, a dataset comprising approximately six million text-image pairs derived from the English subset of LAION-5B. The dataset is designed to support the training and fine-tuning of Face Identity customization models such as InstantID and IP-Adapter. The authors describe a four-stage data filtering pipeline involving resolution and quality checks, face detection ensuring that detected faces occupy at least four percent of the total image area and excluding images with more than three faces, keyword-based text filtering using prompts generated by GPT-4o, and Named Entity Recognition with spaCy to identify text containing personal references.

The authors claim that models fine-tuned on FaceID-6M achieve slightly higher performance compared to proprietary baselines, reporting a gain of +0.04 in the Face Sim metric on a subset of COCO2017 (N = 500) and a +0.13 increase in human evaluation scores (4.26 -> 4.39 on a five-point scale). Scaling experiments show incremental improvements as dataset size increases, with diminishing returns beyond four million samples. The ablation analysis attributes most performance benefits to the presence of faces in images, while other filtering criteria contribute only marginally.

In essence, the paper positions FaceID-6M as the first open, large-scale dataset for FaceID customization that achieves performance comparable to existing industrial resources. It aims to demonstrate that high-quality, identity-preserving generation can be achieved using openly available data. However, the methodological novelty is limited, the reported performance gains are minimal and statistically unsubstantiated, and the dataset construction largely reuses standard filtering heuristics already common in large-scale image-text curation pipelines.

**Strengths:**

- While not conceptually groundbreaking, the paper’s originality lies in its practical aim to provide an open, large-scale dataset for FaceID customization. It combines existing components such as LAION-5B filtering,NER, and GPT-generated keyword expansion into a single, reproducible data pipeline.

- The paper is generally well structured and transparent about its dataset construction process. The methodology is clearly described, with detailed criteria for image selection, face detection thresholds, and text filtering. The inclusion of ablation and scaling studies, even if limited in scope, reflects an effort toward empirical validation.

- The manuscript is clearly written and easy to follow. The multi-step pipeline is explained systematically, and figures and tables effectively communicate the filtering logic and dataset characteristics. The work provides enough implementation detail for partial reproducibility.

- The dataset, if released as stated, could lower the entry barrier for researchers without access to proprietary resources. It offers a potential benchmark for training and evaluating identity-preserving diffusion models such as InstantID and IP-Adapter, thereby contributing to community reproducibility and model auditing.

- The focus on open-source release and reproducibility aligns with growing community priorities in generative AI and facial identity research, addressing concerns about accessibility and replicability in large-scale model training.

**Weaknesses:**

Weaknesses
- The dataset construction approach is largely an incremental application of existing large-scale filtering techniques previously introduced for LAION-derived datasets [1,2]. Similar open-source initiatives, such as CelebA-HQ [3] and FFHQ [4], have already demonstrated principled approaches for curating high-quality, identity-consistent face datasets, including manual verification and bias analysis. In contrast, FaceID-6M relies exclusively on automated heuristics without introducing new filtering principles or supervision mechanisms that could substantively improve data fidelity or fairness.
- The manuscript frames FaceID-6M as the "first open-source" FaceID dataset but fails to engage with concurrent efforts like [5, 10], which explicitly target identity-preserving generation with structured annotations.
-  The choice of a minimum face area of 4 percent and a maximum of three faces per image appears ad hoc and lacks sensitivity analysis. Established face dataset construction work, such as MegaFace [6], evaluates detection thresholds empirically to maintain consistent feature resolution across diverse datasets. The absence of any quantitative justification or performance degradation analysis for alternative thresholds reduces confidence in the dataset’s internal validity.
-  Reported improvements of +0.04 in Face Sim and +0.2 in human evaluation are too small to be meaningful without statistical testing. There are no standard deviations, confidence intervals, or significance tests. Similar studies, for example StyleGAN2’s evaluation of perceptual improvements [7], employed multiple randomized trials and Fréchet Inception Distance (FID) variance analyses to establish statistical robustness.
-  Testing on a manually selected subset of 500 COCO images and 50 Unsplash images introduces sampling bias and prevents generalization. Current dataset papers typically validate across diverse benchmarks such as CelebA-HQ, LFW, and VGGFace2 [3,4,8], enabling comparison to prior art and ensuring cross-domain representativeness. The restricted and hand-picked evaluation here cannot substantiate broad claims about the dataset’s utility.
-  The dataset inherits the demographic imbalances of LAION-5B but does not attempt to quantify or mitigate bias. Prior work has shown that facial generation datasets can reproduce societal stereotypes unless explicitly audited for demographic representation [9]. The lack of demographic transparency conflicts with modern standards for responsible data curation.

The work could improve by grounding its filtering design in empirical analysis, expanding evaluation to established face recognition benchmarks, incorporating fairness audits, providing reproducible filtering artifacts, and contextualizing its contribution relative to concurrent identity-preserving dataset initiatives. Without these revisions, the dataset remains an engineering-scale replication of LAION filtering rather than a scientifically novel contribution.


References

[1] Schuhmann C, Beaumont R, Vencu R, Gordon C, Wightman R, Cherti M, Coombes T, Katta A, Mullis C, Wortsman M, Schramowski P. Laion-5b: An open large-scale dataset for training next generation image-text models. Advances in neural information processing systems. 2022 Dec 6;35:25278-94.

[2] Changpinyo S, Sharma P, Ding N, Soricut R. Conceptual 12m: Pushing web-scale image-text pre-training to recognize long-tail visual concepts. InProceedings of the IEEE/CVF conference on computer vision and pattern recognition 2021 (pp. 3558-3568).

[3] Liu Z, Luo P, Wang X, Tang X. Deep learning face attributes in the wild. InProceedings of the IEEE international conference on computer vision 2015 (pp. 3730-3738).

[4] Karras T, Laine S, Aila T. A style-based generator architecture for generative adversarial networks. InProceedings of the IEEE/CVF conference on computer vision and pattern recognition 2019 (pp. 4401-4410). https://github.com/NVlabs/ffhq-dataset

[5] Xu J, Li S, Wu J, Xiong M, Deng A, Ji J, Huang Y, Mu G, Feng W, Ding S, Hooi B. $\text {ID}^ 3$: Identity-Preserving-yet-Diversified Diffusion Models for Synthetic Face Recognition. Advances in Neural Information Processing Systems. 2024 Dec 16;37:77777-98.

[6] Kemelmacher-Shlizerman I, Seitz SM, Miller D, Brossard E. The megaface benchmark: 1 million faces for recognition at scale. InProceedings of the IEEE conference on computer vision and pattern recognition 2016 (pp. 4873-4882).

[7] Karras T, Laine S, Aittala M, Hellsten J, Lehtinen J, Aila T. Analyzing and improving the image quality of stylegan. In Proceedings of the IEEE/CVF conference on computer vision and pattern recognition 2020 (pp. 8110-8119).

[8] Cao Q, Shen L, Xie W, Parkhi OM, Zisserman A. Vggface2: A dataset for recognising faces across pose and age. In2018 13th IEEE international conference on automatic face & gesture recognition (FG 2018) 2018 May 15 (pp. 67-74). IEEE.

[9] Buolamwini J, Gebru T. Gender shades: Intersectional accuracy disparities in commercial gender classification. InConference on fairness, accountability and transparency 2018 Jan 21 (pp. 77-91). PMLR.

[10] Chen Z, Sun K, Zhou Z, Lin X, Sun X, Cao L, Ji R. Diffusionface: Towards a comprehensive dataset for diffusion-based face forgery analysis. arXiv preprint arXiv:2403.18471. 2024 Mar 27.

**Questions:**

1. How were the 500 COCO and 50 Unsplash samples chosen, and are the small reported gains statistically significant?

 2. What evidence supports the 4% face-size threshold and the three-face limit?

 3. Why are other recent open datasets for identity preservation not used for comparison?

 4. Can the authors publish the exact GPT-4o prompts and keyword lists for reproducibility?

 5. Has any demographic or bias analysis been conducted to assess fairness and representativeness?

 6. Under what license and consent framework will the dataset be released, and how are copyright or GDPR issues addressed?

 8. Why do performance gains plateau beyond four million samples, and is the additional data volume justified?

---

### Official Review · Reviewer_vRFc · 2025-10-28

**Soundness:** 2
**Presentation:** 3
**Contribution:** 2
**Rating:** 4
**Confidence:** 2

**Summary:**

This paper introduces FaceID-6M, a large-scale, open-source dataset of 6 million high-quality text-image pairs. High-quality images were selected using Language Filtering, Image Filtering, and Text Filtering. Experiments demonstrate that models trained on FaceID-6M achieve performance that is comparable to currently available industrial models.

**Strengths:**

1. The fundamental idea of this paper is technically correct.
2. This dataset addresses a need for a large-scale, open-source benchmark for FaceID customization research.
2. The paper is well written and easy to follow.

**Weaknesses:**

1. While the paper introduces a valuable and well-constructed dataset, the technical contribution beyond data collection and filtering is limited.
2. The paper does not provide a demographic analysis of the FaceID-6M dataset, such as the distribution of gender, ethnicity, or age. Presenting this information is crucial, as a diverse and well-balanced dataset is essential for training robust and fair models, minimizing potential biases.
3. In Table 2, the authors demonstrate the impact of applying Resolution Constraints and Minimum Face Size Constraints. However, the specific threshold values chosen (e.g., 512 pixels for resolution and a 4% face-to-image area ratio) are presented without experimental verification. It would be more convincing if the authors included experiments to show how different threshold values affect the final model performance.
4. A significant portion of Section 4, "FaceID Customization Based on FaceID-6M," is dedicated to describing existing methods. While this provides useful context, it largely reiterates the work of others (e.g., IP-Adapter, InstantID) without introducing technical innovation from the authors themselves.
5. In the Image Filtering stage, the authors chose to remove images containing more than three faces.  While this likely purifies the data for single-subject tasks, retaining these multi-person images could have offered a more realistic dataset. Could the authors elaborate on this design choice?

**Questions:**

The technical contribution beyond data curation is minimal, lacking demographic analysis of the FaceID-6M dataset's distribution across gender, ethnicity, and age. The filtering thresholds (512 pixels resolution, 4% face-to-image ratio) lack experimental justification. Additionally, the decision to exclude multi-person images may limit the dataset's real-world applicability.

---

### Official Review · Reviewer_pHtQ · 2025-10-31

**Soundness:** 2
**Presentation:** 2
**Contribution:** 1
**Rating:** 2
**Confidence:** 4

**Summary:**

The paper proposes FaceID-6M, a large-scale and open-source dataset for FaceID customization training. The authors constructed the dataset by filtering the LAION-5B dataset to select image-text pairs that contained high-quality face images and meaningful English descriptions. IP-Adapter and InstantID models trained on FaceID-6M provide better generation quality than their original models.

**Strengths:**

IP-Adapter and InstantID models trained on FaceID-6M provide better generation quality than their original models, verified by both quantitative, qualitative, and human evaluation.

**Weaknesses:**

- The paper claims to release the first large-scale, open-source FaceID dataset containing high-quality text-image pairs. This claim is incorrect. Many similar datasets [1,2,3,4] were released before this work.
- The authors should provide important statistics of the FaceID-6M: the number of images, the average number of faces per image, the number of distinct identities (if possible to count), gender/ethnicity/age distribution.
- I recommend to downgrade Language Filtering to be a side note rather than one of the main data filtering steps. I know the reason but English-only sounds biased and undesirable.
- L166: I cannot find the detection model named "Antelopev2" in the InsightFace repo.
- The authors remove images that have more than 3 faces. Given the identity leakage issue already appears when having two or three faces, the reason for this decision needs more clarification.
- Other important factors in facial image filterring should be considerred: head pose, occlusion, expression, and image quality.
- Sections 4.2.1 and 4.2.2 are lengthy and redundant. The authors can summarize these sections as they train IP-Adapter and InstantID on the FaceID-6M dataset following their training protocols described in the original papers.
- Table 1: What does "Fine-tuned on Laion-FaceID" mean? Is "Laion-FaceID" FaceID-6M? Also, were the models trained from-scratch or finetuned from the public pretrained checkpoints?
- Section 5.4 mentions Figure 4, which is in the Appendix. The authors should move that figure to the main text.

[1]. https://github.com/ddw2AIGROUP2CQUPT/Large-Scale-Multimodal-Face-Datasets.
[2]. Yu, Jianhui, et al. "Celebv-text: A large-scale facial text-video dataset." In CVPR 2023.
[3]. https://github.com/Yutong-Zhou-cv/FFHQ-Text_Dataset
[4]. https://github.com/FacePerceiver/LAION-Face

**Questions:**

- The paper claims to release the first large-scale, open-source FaceID dataset containing high-quality text-image pairs. This claim is incorrect. Many similar datasets [1,2,3,4] were released before this work.
- The authors should provide important statistics of the FaceID-6M: the number of images, the average number of faces per image, the number of distinct identities (if possible to count), gender/ethnicity/age distribution.
- L166: I cannot find the detection model named "Antelopev2" in the InsightFace repo.
- The authors remove images that have more than 3 faces. Given the identity leakage issue already appears when having two or three faces, the reason for this decision needs more clarification.
- Other important factors in facial image filterring should be considerred: head pose, occlusion, expression, and image quality.
- Table 1: What does "Fine-tuned on Laion-FaceID" mean? Is "Laion-FaceID" FaceID-6M? Also, were the models trained from-scratch or finetuned from the public pretrained checkpoints?

**Details Of Ethics Concerns:**

The dataset contains human faces and identity information like subject names, job, gender, and ethnicity. Although these data already exist in the LAION dataset, and this work filters that set, ethics review is still needed.

---

### Meta-Review · Area_Chair_od2i · 2026-01-08

**Summary:**

The decision was informed by converging concerns about limited novelty, weak empirical justification, and ethical risk. Specifically, Multiple reviewers argued that the dataset curation is largely incremental, reusing standard LAION-style filtering without new principles. Moreover, reported gains are small and statistically unsupported, evaluated on narrow, hand-picked benchmarks. Last, demographic audits, bias analysis, and privacy/GDPR safeguards are missing, raising serious ethical concerns for releasing a large face dataset.

**Reviewer Concerns:**

**Addressed:**
* The presentation issues, such as naming confusions and implementation details.
* Empirical benefit of FaceID-6M for training, quantitative/qualitative analysis, and human studies.

**Outstanding:**
* Limited scientific contribution beyond curation
* Overstated novelty / incorrect “first open-source” claim
* Lack of demographic and bias analysis
* Weak statistical rigor and narrow evaluation

**Reviewer Scores:**

Two reviewers pHtQ and c4sY are unlikely to change their ratings, and they may stick with the "solid reject." While vRFc felt positive about this work, the reviewer flags limited technical contribution, a missing demographic analysis, and unjustified thresholds. The rebuttal did not fully address these concerns, and a full flip to "accept" is unlikely. NiYi may retain the positive rating but may be affected by the panel's overall ratings and lean to the negative side.

---

### Decision · Program_Chairs · 2026-01-26

Reject